# Axillary Lymph Node Metastasis from Ovarian Carcinoma: A Systematic Review of the Literature

**DOI:** 10.3390/jpm13111532

**Published:** 2023-10-26

**Authors:** Nektarios Koufopoulos, Abraham Pouliakis, Ioannis Boutas, Menelaos G. Samaras, Adamantia Kontogeorgi, Dionysios Dimas, Kyparissia Sitara, Andriani Zacharatou, Magda Zanelli, Andrea Palicelli

**Affiliations:** 1Second Department of Pathology, Medical School, National and Kapodistrian University of Athens, Attikon University Hospital, 12462 Athens, Greece; apouliak@med.uoa.gr (A.P.); menelaos.g.samaras@gmail.com (M.G.S.); anniezacharatoumed25@gmail.com (A.Z.); 2Breast Unit, Rea Maternity Hospital, Palaio Faliro, 17564 Athens, Greece; iboutas@med.uoa.gr; 33rd Department of Obstetrics and Gynecology, National and Kapodistrian University of Athens, Attikon University Hospital, 12462 Athens, Greece; mandokont@med.uoa.gr; 4Breast Unit, Athens Medical Center, Psychiko Clinic, 11525 Athens, Greece; diodimas@med.uoa.gr; 5Department of Internal Medicine, “Elpis” General Hospital of Athens, 11522 Athens, Greece; kdsitar@med.uoa.gr; 6Pathology Unit, Azienda USL-IRCCS di Reggio Emilia, 42123 Reggio Emilia, Italy; magda.zanelli@ausl.re.it

**Keywords:** ovarian carcinoma, serous carcinoma, axillary lymph node metastasis, supradiaphragmatic ovarian metastasis, stage IV ovarian cancer

## Abstract

Background: Axillary lymph node metastasis is a rare stage IV ovarian carcinoma manifestation. This manuscript aims to systematically review the literature regarding axillary lymph node metastasis from ovarian carcinoma. Methods: We searched three medical internet databases (PubMed, Scopus, and Web of Science) for relevant articles published until 22 July 2023. Cases describing supraclavicular or intramammary lymph node metastases and concurrent metastasis to the breast were excluded. Results: After applying eligibility/inclusion and exclusion criteria, twenty-one manuscripts describing twenty-five cases were included from the English literature. Data were collected and analyzed regarding demographic, clinical, laboratory, radiological, histopathological, and oncological characteristics. Conclusions: We analyzed the clinical and oncological characteristics of patients with axillary lymph node metastasis from ovarian carcinoma, presented either as an initial diagnosis of the disease or as a recurrent disease. The analysis we performed showed a significant difference only in the serum CA-125 level (*p* = 0.004) between the two groups. There was no observed difference in womens’ survival.

## 1. Introduction

Ovarian cancer ranks fifth in deaths caused by gynecological cancers [1]. A lack of specific signs and symptoms contributes to late diagnosis, with most cases presenting in the advanced stage [2,3]. Usual routes of spread are by peritoneal seeding and lymphatic infiltration, with the most common metastatic sites being nearby organs (uterus, bladder, sigmoid colon, or the rectum) and pelvic and para-aortic lymph nodes. In advanced stages, metastatic disease can be found in pleural effusion, liver, spleen, lungs, bones, or extra-abdominal lymph nodes [4].

By definition, an axillary metastasis from ovarian cancer corresponds to FIGO Stage IVb. Patients with axillary lymph node metastasis usually present in two different scenarios. Indeed, the diagnosis of ovarian cancer stage IV is given at the first presentation after investigating abnormal lymphadenopathy, or lymph node metastasis is found during clinical or radiological follow-up examination as a recurrence of the disease with distant metastasis in patients already treated for ovarian cancer. Most cases of axillary lymph node metastases from ovarian carcinoma reported in the English literature are found to accompany breast metastases [5]. Only a few cases of isolated axillary lymph node metastasis from ovarian cancer have been reported. The first such case was reported in 1997 with another 24 following until July 2023.

This article aims to systematically review the literature regarding cases with axillary lymph node metastasis from ovarian carcinoma without synchronous breast metastases. We present the clinical and oncological characteristics of patients with axillary lymph node metastasis from ovarian carcinoma presented either as an initial diagnosis or as a recurrent disease. We focus on the diagnostic challenges of these cases and describe the treatment modalities provided in each case concerning survival.

## 2. Materials and Methods

### 2.1. Systematic Review

The systematic review of the literature was performed according to the “Preferred Reporting Items for Systematic Reviews and Meta-Analyses” (PRISMA) guidelines (http://www.prismastatement.org/; accessed on 22 July 2023) (Figure 1) to identify published cases of axillary lymph node metastasis from ovarian carcinoma. This study has not been registered in PROSPERO or any other relevant databases.

Our retrospective observational study search was conducted through the PICO process:**P**opulation: Women with a diagnosis of primary ovarian cancer and axillary lymph node metastases without synchronous metastatic disease to the breast;**I**ntervention: Surgical treatment of the primary ovarian tumor;**C**omparison: None;**O**utcome: Patient treatment and follow-up.

We searched for ((“Axillary”) AND (“Lymph Node” OR “Lymph Nodes”) AND (“Metastasis”) AND (“Ovarian” AND (“Cancer” OR “Carcinoma”)) in the PubMed (all fields; 89 results; https://pubmed.ncbi.nlm.nih.gov, accessed on 22 July 2023), Scopus (Title/Abstract/Keywords; 221 results; https://www.scopus.com/, accessed on 22 July 2023), and Web of Science (all fields, 107 results; https://login.webofknowledge.com, accessed on 22 July 2023) databases. No time limitation was set in our search. No additional limitations or filters were set while performing the search. The following criteria were applied:
Eligibility/Inclusion Criteria:Study design: We only included primary studies reporting human cases of axillary lymph node metastasis in patients with ovarian carcinoma without synchronous primary or metastatic tumors localized in the breast.Population: We included studies that involved adult patients diagnosed with ovarian carcinoma and provided information on axillary lymph node involvement or metastasis.Intervention or exposure: We included studies that examined any treatment or intervention for ovarian carcinoma, including surgery, chemotherapy, radiation therapy, or targeted therapies.Outcome: We included studies that reported on the presence or absence of axillary lymph node metastasis as an outcome measure.Language: The included studies were restricted to the English language.Exclusion Criteria:Review articles and editorials: We excluded narrative reviews, systematic reviews, meta-analyses, opinion pieces, and other articles that did not present original research findings.Studies with breast or intramammary lymph node involvement: We excluded studies with breast involvement by ovarian carcinoma and axillary lymph node metastases or metastasis to intramammary lymph nodes.Studies with inadequate/aggregated data: We excluded studies with incomplete or insufficient/aggregated data.Irrelevant studies: We excluded studies that did not address axillary lymph node involvement or metastasis in patients with ovarian carcinoma.

Three authors (IB, DD, and AK) worked independently to remove any duplicate papers and reviewed titles and abstracts of all the results the searches yielded (*n* = 417). After applying all inclusion and exclusion criteria, only 19 articles were deemed eligible for further screening. These papers were obtained in full-text format, and the snowball procedure was followed for any additional relevant references. For any disagreements in the selection process, consensus between reviewers was used to deem a study worthy of inclusion. After the review of full texts, 21 articles reporting 25 cases were selected for data extraction [6,7,8,9,10,11,12,13,14,15,16,17,18,19,20,21,22,23,24,25,26]. The data collection that followed was case- and study-related.

### 2.2. Statistical Analysis

Available study data, despite being limited, was used to perform statistical analyses. This was performed via the SPSS for Windows version 28.0.1 software platform (IBM Inc, Armonk, NY, USA). Descriptive values were expressed as mean ± standard deviation (SD) as well as with median value and the first and third quartiles (Q1–Q3); for the categorical data, the frequencies and percentages are presented. As normality was not possible to be ensured via the Shapiro–Wilk test, comparisons of the arithmetic data (woman’s age, CA-125 levels) between groups were performed via non-parametric tests, specifically the Mann–Whitney U test. Comparisons between groups for the categorical data were performed by using Fisher’s exact test. Furthermore, survival/follow-up data were available for numerous cases; therefore, via the Cox regression approach, it was possible to compare the survival outcome of lymph node metastases diagnosed at first presentation with that of cases presenting as a recurrence. The significance level (*p*-value) was set to α = 0.05 and all tests were two-sided.

## 3. Results

### 3.1. Demographic and Clinical Data of All Cases

The median age at presentation was 58 years (mean ± SD: 57.3 ± 12.8, range: 37–79). Regarding laterality, fourteen cases presented metastasis in the right axillary region, five in the left, and two had bilateral axillary lymph node metastases.

Twenty-one cases reported serum CA-125 levels at the time of the axillary lymph node diagnosis; thirteen cases reported biomarker levels in U/mL, two in ng/mL, two in μm/mL, one in U/L, and three presented no measurement units. Considering levels measured in U/mL, the median CA-125 levels at presentation were 437 U/mL (median: 433.04 U/mL, range: 26.7–1024 U/mL).

Thirteen cases manifested the axillary lymph node metastasis as an initial presentation of the disease, either as the first symptom that led to the diagnosis or as lymph node enlargement through image study during the work-up for an ovarian mass [6,7,9,14,15,17,18,20,22,23,24]. Twelve cases exhibited lymph node metastasis as a recurrence of a previously treated ovarian carcinoma [8,10,11,12,13,16,19,21,23,25,26].

### 3.2. Cases of Axillary Metastasis as the Initial Presentation

#### 3.2.1. Demographic and Clinical Data

The median age of the cases that presented with axillary lymph node metastasis during the first diagnosis was 55 years (mean: 57.2 ± 12.0, range: 39–78).

Clinically, three cases presented with abnormal findings in breast screening evaluation [6,15,17]. Four presented with enlargement in the axillary region [9,14,20,24]. One patient presented with bilateral mastitis [7]. Six cases were diagnosed with abnormal enlargement in the axillary region during clinical or imaging examination for a newly diagnosed ovarian mass or ascites after symptoms of abdominal distension or bloating [18,22,23]. In one of the two cases reported by Eitan et al. [22], ovarian carcinoma was diagnosed after examination of persistent umbilical purulent discharge. In one of the three cases reported by Zuhdy et al. [23], the abnormal axillary lymph node enlargement was discovered during the work-up for a papillary serous cystadenocarcinoma diagnosed after an ovarian cystectomy.

Seven cases reported the size of the adnexal mass, with the median size being 6 cm (mean: 7.4 cm, range: 3–14 cm). Regarding laterality, all cases reported the side of the adnexal mass in the imaging study. Five were right, two left, and six bilateral. Eight cases presented with right, four with left, and one presented as bilateral axillary lymph node involvement.

All cases reported CA-125 levels at the time of axillary lymph node diagnosis; eight reported biomarker levels in U/mL, one in μm/mL, one in ng/mL, and three presented no measurement units. Considering the biomarker levels expressed in U/mL, the median CA-125 units at presentation were 660.9 U/mL (mean: 617.4 ± 276.4 U/mL, range: 230 U/mL-1024 U/mL). The summary of the clinicopathological characteristics is displayed in Table 1.

#### 3.2.2. Diagnostic Workup

A pelvic imaging study performed during the investigation of these cases showed the presence of ovarian masses without further abnormal findings in two patients [6,14]. In eight cases, there was peritoneal and/or omental involvement in addition to [15,17,18,22,23] or without [7,22] an ovarian tumor; five cases reported the presence of ascites [7,17,18,23,24], while retroperitoneal lymph node enlargement was found in three patients [20,22,23]. In the case reported by Moustafa and Benyon [9], ovarian and para-aortic lymph node enlargement was noted. In one case, an ovarian mass and ascites were found [24], and one reported the presence of pleural effusion [18].

All cases were subjected to axillary lymph node aspiration or biopsy. In 3/5 (60%) cases that were subjected only to fine-needle aspiration cytology examination (FNA), the pathologist was able to diagnose a metastatic adenocarcinoma but the primary tumor origin was not identified [7,9,20]; in the remaining two cases (40%), the pathologist was able to diagnose a metastatic ovarian carcinoma [23]. Five cases were subjected to lymph node biopsy. Two identified metastatic adenocarcinoma cells without commenting on origin [6,22] and three identified metastasis from ovarian origin [18,22,24]. Three cases were subjected to both FNA and biopsy of the lymph node. In all cases, FNA cytology could identify a metastatic adenocarcinoma from an undefined primary origin, and biopsy provided the information regarding ovarian origin [14,15,17]. In three instances, cytological examination of the ascitic fluid revealed the presence of malignancy [17,20] or metastatic carcinoma [18].

Regarding the histopathological examination of the surgical specimen, all cases except one [6] were diagnosed as high-grade serous carcinoma using different terminology, such as serous adenocarcinoma in four patients [9,20,22], papillary serous cystadenocarcinoma in three [17,23], papillary serous carcinoma of the ovary in two [7,18], and serous cystadenocarcinoma in two [14,24]. The case reported by Hockstein et al. [6] was initially considered breast carcinoma with lymph node and ovarian metastasis despite the absence of a primary breast tumor. After recurrence to the colon and a more elaborate immunohistochemical study, the diagnosis of poorly differentiated ovarian adenocarcinoma was made. In one of the two cases reported by Eitan et al. [22], the histopathological examination of the surgical specimen revealed a fallopian tube involvement consistent with serous tubal intraepithelial carcinoma.

#### 3.2.3. Surgical–Oncological Management

All patients except one [15] proceeded with surgical treatment. Seven patients underwent cytoreductive surgery as the first line of treatment, followed by systemic chemotherapy [7,9,14,17,20,23] or hormonal therapy [6]; three cases were treated with neoadjuvant chemotherapy and interval debulking followed by adjuvant chemotherapy [22,23]. One case was treated with neoadjuvant chemotherapy and cytoreductive surgery [18]. After optimal cytoreduction, two patients were subjected to hyperthermic intraperitoneal chemotherapy (HIPEC) with cisplatin [14,17]. In one case, rectosigmoidectomy was performed after the exploratory laparotomy revealed penetration from ovarian carcinoma. Three patients underwent axillary clearance during cytoreductive surgery [7,20,23].

Regarding chemotherapy, eleven cases were reported on specific regimens. The carboplatin–paclitaxel regimen was used in eight patients either as neoadjuvant chemotherapy, adjuvant treatment, or both. One case received a combination of cisplatin, epirubicin, and cyclophosphamide [7]. Three patients [22,24] received bevacizumab in combination with carboplatin–paclitaxel as adjuvant chemotherapy. Adjuvant tamoxifen was administered due to an initial misdiagnosis of metastatic breast carcinoma and later changed into carboplatin–paclitaxel in the case reported by Hockstein et al. [6].

#### 3.2.4. Molecular Studies

The two cases reported by Eitan et al. [22] were tested for *BRCA1* and *BRCA2* mutations. One of the two patients was found positive for a mutation in *BRCA1* (5382insC); the patient had a noncontributory medical and family history and was 51 years old at diagnosis of high-grade serous carcinoma with umbilical skin and left axillary lymph node metastases. In the second patient, the testing for *BRCA1* and *BRCA2* mutations was negative.

#### 3.2.5. Follow-Up

Nine cases provided information regarding follow-up. Two patients were lost to follow-up [23], and one more case did not provide relevant details [18]. Five reported no disease recurrence [7,14,17,20,24] with a median disease-free survival (DFS) of 12 months (mean: 33.6, range: 12–84). Two cases recurred with pelvic, peritoneal, mediastinal, retroperitoneal lymph node, and bilateral axillary lymph node involvement in a period ranging from six to seven months after chemotherapy [22]. The longest DFS (7 years) was reported by Sibio et al. [17] after optimal cytoreduction, followed by HIPEC with cisplatin and adjuvant chemotherapy with six courses of carboplatin–paclitaxel in a 49-year-old patient. In the case reported by Hockstein et al. [6], the patient was treated for metastatic breast carcinoma by administering adjuvant tamoxifen after optimal cytoreductive surgery and developed pelvic recurrence one year postoperatively. The patient was then treated for metastatic ovarian carcinoma; the regimen was changed to adjuvant carboplatin–paclitaxel after the excision of the mass and remained disease-free for eight months after completing six courses of chemotherapy. The summary of the treatment and follow-up features is displayed in Table 2.

### 3.3. Cases of Axillary Metastasis as Recurrence

#### 3.3.1. Demographic and Clinical Data

The twelve cases presenting axillary lymph node metastasis as a disease recurrence of a previously treated ovarian carcinoma had a median age at presentation of 60.5 years (mean ± SD: 57.4 ± 14.2, range: 37–79).

Ten cases reported the period between the previously treated ovarian carcinoma and the axillary recurrence. The median interval was 26.5 months (average: 26.4, range: 6–48).

Only four cases reported the laterality of previous ovarian involvement, with two displaying right disease [13,21] and another two [23,26] bilateral disease. In contrast, all cases reported laterality in the axilla. Seven patients presented with metastasis in the right axilla [10,11,12,13,21,23], three in the left [16,25,26], and two with bilateral [8,19] axillary lymph node involvement.

Four asymptomatic patients discovered the axillary lymph node enlargement during regular follow-up examination for previously treated ovarian cancer [8,13,16,23]. Seven cases presented with symptomatic enlargement in the axillary region [10,11,12,19,21,25].

Eight cases reported CA-125 levels at the time of axillary lymph node diagnosis. Five reported biomarker levels in U/mL, one in ng/mL, one in μm/mL, and one in U/l. In cases reporting biomarker values in U/mL, the median CA-125 was 104 U/mL (mean: 164.5 ± 166.8 U/mL, range: 26.7–437 U/mL). The summary of the clinical and demographic characteristics is displayed in Table 3.

#### 3.3.2. Initial Diagnosis and Management

The initial histological diagnosis was high-grade serous carcinoma in eight cases reported as papillary serous carcinoma in four (50%) [8,10,12,19] and serous carcinoma in four (50%) [16,21,25,26]. One patient was diagnosed with epithelial ovarian carcinoma [10] and two with ovarian carcinoma [11,13]. In one case, the initial diagnosis was carcinosarcoma with heterologous elements consisting of papillary serous carcinoma and chondrosarcomatous differentiation [23].

In all cases, stage III was assigned, IIIA in two [10], IIIB in two [11,23], and stage IIIC in six [8,12,13,16,19,21,25,26].

Nine patients underwent primary debulking surgery and adjuvant chemotherapy [8,10,11,12,13,16,21,25,26]. The carboplatin–paclitaxel regimen was administered in seven cases [8,10,12,13,16,25,26] and taxane with paclitaxel in one [21]. The ninth case did not provide further details concerning adjuvant treatment [11]. Three patients were treated with neoadjuvant chemotherapy consisting of carboplatin–paclitaxel, interval debulking, and adjuvant chemotherapy [10,19,23].

#### 3.3.3. Diagnostic Work-Up of Recurrence

In all cases, imaging studies confirmed the presence of an axillary mass. In one of the two cases reported by Ozmen et al. [10], a mass lesion located at the right lateral wall of the rectum was found on abdominal computed tomography (CT) scans. Subsequent biopsies revealed metastases from ovarian carcinoma.

In seven cases, cytological or histopathological lymph node examination via FNA or biopsy was performed. In four patients, both procedures were performed. In three cases, the diagnosis of metastasis from ovarian carcinoma was made [8,11,21], and in one, no further defined malignant cells were found [13]. In two cases, lymph node biopsy revealed a metastasis from ovarian carcinoma [10] or a serous carcinoma [26]. Two patients underwent cytological and histopathological examinations revealing metastatic adenocarcinoma [12] and metastatic ovarian carcinoma [19], respectively.

The final histological diagnosis was metastatic serous carcinoma in six cases, reported as serous papillary adenocarcinoma in two [10,12], serous carcinoma in three [13,16,19], and papillary carcinoma [10] in one. In three cases, the final diagnosis was metastatic ovarian carcinoma [8,11,23].

#### 3.3.4. Surgical–Oncological Management of Recurrence

Five cases proceeded with excisional biopsy of the enlarged axillary lymph node [11,13,16,23,25], while four proceeded with axillary lymph node dissection of the affected side [8,10,12]. One case did not provide details concerning treatment [19]. In the case reported by Nazos et al. [21], complete cytoreduction was not feasible. In one case, concurrent rectal metastasis was managed with low anterior resection [10]. In the case of Orris et al. [8], lymph node dissection, omentectomy, and peritoneal biopsies were performed in a second-look operation.

Eight cases provided information on adjuvant therapy. Six patients were treated with a combination of carboplatin–paclitaxel [8,10,13,23,25,26]. In one case, adjuvant treatment was not specified [10]. The patient reported by Nazos et al. [21] received only radiotherapy. Four cases did not mention further management [11,12,16,19]. In two instances, metastasis to the axilla was preceded by recurrence to other organs. In the case reported by Orris et al. [8], disease recurrence was detected during a second look operation in the right cul-de-sac, and intraperitoneal cisplatin therapy was administered through a Tenckhoff catheter. In the case reported by Woo and Kim [12], a CA-125 elevation at 45 months was considered disease recurrence despite the absence of findings on imaging studies. The patient received a regimen consisting of paclitaxel, carboplatin, and gemcitabine. The summary of the treatment and follow-up information is displayed in Table 4.

#### 3.3.5. Molecular Studies

Two patients with noncontributory medical and family history were tested for *BRCA1* and *BRCA2* mutations in [25,26]. No pathogenic mutation was detected in either case.

#### 3.3.6. Follow-Up

For six cases, information concerning patient follow-up was available. Five reported no recurrence of the disease with a median DFS of 12 months (mean: 18, range: 6–36) [10,13,16]. One case reported the development of peritoneal recurrence 22 months after the axillary lymph node diagnosis, which was treated with excisional biopsy and six cycles of carboplatin–paclitaxel 50 months after primary surgery. The patient was subsequently treated with chemotherapy (six cycles of carboplatin–paclitaxel regimen) achieving a stationary disease course; the regimen was later shifted into three cycles of paclitaxel–gemcitabine, followed by aromatase inhibitor, resulting in a stable disease status 16 months after the diagnosis of peritoneal recurrent disease [23]. The longest DFS (3 years) was reported by Goyal et al. in a 68-year-old patient after an excisional biopsy of the affected axillary lymph node and without further treatment [16].

### 3.4. Comparison of Cases with Initial Diagnosis and Recurrence

Comparing the age of patients with axillary lymph node metastases at initial disease presentation (group 1) with that of the women showing axillary nodal metastases as disease recurrence (group 2), no significant statistical difference was found between the two groups (*p* = 0.979). Further statistical analysis of the two groups was not able to prove any statistical significance regarding the side of ovarian involvement (left, right, or bilateral), or the lymphatic node metastasis location (*p* > 0.999 for both tests). In relation to CA-125 levels, we compared them when reported in the same units (U/mL); we found that group 1 women had higher median value (660.9 U/mL; Q1-Q3: 364–841 U/mL) than group 2 patients (104 U/mL; Q1-Q3: 38.9–320.5 U/mL) (*p* = 0.004), proving around four times lower levels of CA-125 on recurrence. Finally, using the available data known for patients alive without disease or alive with disease, no difference in survival was found (*p* = 0.771). The relevant survival curves appear in Figure 2.

## 4. Discussion

Breast carcinoma is the most common solid tumor metastasizing to the axillary lymph nodes. Less commonly, carcinomas of the lung [27], thyroid [28], stomach [29], esophagus [30], and endometrium [31] may present with axillary metastases.

Isolated axillary lymph node metastasis constitutes a rare presentation of ovarian carcinoma spread. It usually coincides with breast or other supradiaphragmatic lymph node metastases [32,33,34]. The occurrence of enlarged lymph nodes in the absence of mammographic findings could also be due to non-Hodgkin or Hodgkin lymphoma as well as lymphadenitis caused by several different infectious agents (such as Mycobacterium tuberculosis or other Mycobacteria, Streptococcus, Francisella tularensis, Yersinia pestis, Bartonella, Epstein–Barr virus, Cytomegalovirus, Toxoplasma gondii, Brucella, Treponema pallidum, Histoplasma capsulatum, etc.) [35,36]. Also, fibroadenomas arising in axillary ectopic breast tissue may appear as an enlarged lymph node [37]. Therefore, the clinical differential diagnosis is vague.

The pathological diagnosis of axillary metastasis from ovarian carcinoma can be challenging, mainly when it occurs as the first presentation of the disease. This is highlighted by the fact that the first reported case by Hockstein et al. in 1997 was misdiagnosed as a primary carcinoma of the breast with axillary lymph node metastasis. Lack of clinical suspicion due to the rarity of such cases and the occasional histological similarity of poorly differentiated metastatic mammary and ovarian carcinoma can render the accurate pathological diagnosis problematic. Also, the possibility of axillary metastasis from an occult breast carcinoma cannot be ruled out [38]. Elevated serum CA-125 levels may be a helpful diagnostic clue in the clinical scenario of axillary metastasis as the initial presentation. The use of appropriate immunohistochemical markers may aid in problematic cases. An immunohistochemical panel consisting of PAX-8, WT-1, GATA3, and Mammaglobin can help differentiate breast and ovarian carcinomas in most cases. PAX-8 is a member of the paired box gene family of transcription factors that regulate organogenesis [39]. PAX-8 shows nuclear staining in epithelial neoplasms of the female genital tract (ovaries, endometrium endocervix, and fallopian tube), thyroid, kidney, and thymus [40]. It is expressed in various tumors, including, among others, ovarian serous epithelial tumors [41]. WT-1 is a transcription factor that has an essential role in the normal development of the urogenital system [42]. WT-1 antibody is expressed immunohistochemically in several tumors, among them ovarian serous carcinoma [43]. It should be noted that WT-1 may stain almost two-thirds of mammary mucinous carcinomas [44]. GATA 3 belongs to the GATA family of transcription factors [45] and is expressed mainly in breast carcinoma, urothelial carcinoma, paraganglioma/pheochromocytoma, and squamous cell carcinoma of the skin, and less commonly in a variety of tumors of the kidney, lung, liver, pancreas, stomach, ovaries, endometrium, thyroid, and salivary gland carcinoma [46]; it can be very useful in cases of metastatic breast carcinoma [47]. Mammaglobin is a member of small epithelial secretory proteins of the uteroglobin/Clara cell protein family [48] and displays a high specificity for breast carcinoma [49,50].

In any case, positive staining of PAX-8 and WT-1 antibodies combined with a lack of staining of GATA-3 and mammaglobin should be expected in cases of ovarian serous carcinoma metastasis to axillary lymph nodes [51]. In our experience with a similar case, the morphology (solid nests of atypical cells with significant nuclear pleomorphism, a prominent nucleolus and a high mitotic index, and lack of psammoma bodies) was not helpful to identify the origin of the tumor. Therefore, immunohistochemical stains for PAX-8, WT-1, P53, GATA 3, and mammaglobin were performed. Positive staining for the first three and negative staining for the latter two helped us determine the primary ovarian origin of the tumor (Figure 3).

Extra-abdominal ovarian carcinoma metastasis is uncommon. The rate and location of metastatic spread are stage and histotype-dependent, with serous carcinoma displaying a higher tendency for lymph node involvement than non-serous tumors [52,53]. All the cases of ovarian carcinomas included in our analysis showed a serous histotype. Indeed, high-grade serous carcinoma is the most frequent primary carcinoma of the ovary, typically accounting for most of the deaths from ovarian cancer; while it usually presents at advanced stage with peritoneal involvement, breast or axillary involvement is still infrequent [54].

CA-125 serum levels can be increased in cases of ovarian carcinoma irrelevant of histotype (serous or non-serous) and the decreasing values can be an indication of response to adjuvant therapy [55,56,57]. Regarding the difference in the CA-125 level that proved statistically significantly different among the two groups, we believe that it is due to the higher tumor burden of cases in the first group (initial diagnosis group) compared with the second group.

The mean interval time from the initial presentation of the disease to the axillary lymph node recurrence was reported as 24 months. The mean overall survival was 28.2 months for cases with initial disease presentation and 39.4 months for patients with disease recurrence.

The limitations of our review are the relatively small amount of studies included, the inhomogeneity of cases, and the lack of published data for long-term survival. Regarding the small number of reported cases, we believe that the reason is two-fold. The first is the rarity of the described distant metastases and the second is that there is a global decrease in the number of case reports [58]. On the other hand, our manuscript describes a fairly unknown field of gynecological oncology with no standardized management and treatment. Further research in this field can potentially provide more prolonged survival to oncological patients with stage IV ovarian cancer.

## 5. Conclusions

In this study, we collected data and presented a systematic review of clinical and oncological characteristics of isolated axillary lymph node metastasis from ovarian carcinoma. We divided our data review into cases presented as initial diagnoses of ovarian carcinoma and patients with recurrent disease. Our study does not show a significant difference in survival between the two groups. The sole difference was the higher level of CA-125 in the cases that present as initial diagnosis. To the best of our knowledge, this is the first study thoroughly reviewing axillary lymph node metastasis from primary ovarian cancer; there is a need for more data about this topic and multicentric-focused studies are required.

## Figures and Tables

**Figure 1 jpm-13-01532-f001:**
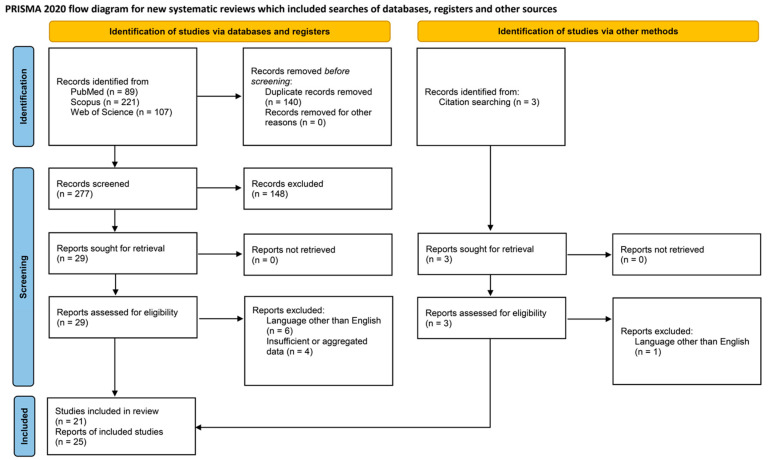
Systematic literature review: PRISMA flow chart.

**Figure 2 jpm-13-01532-f002:**
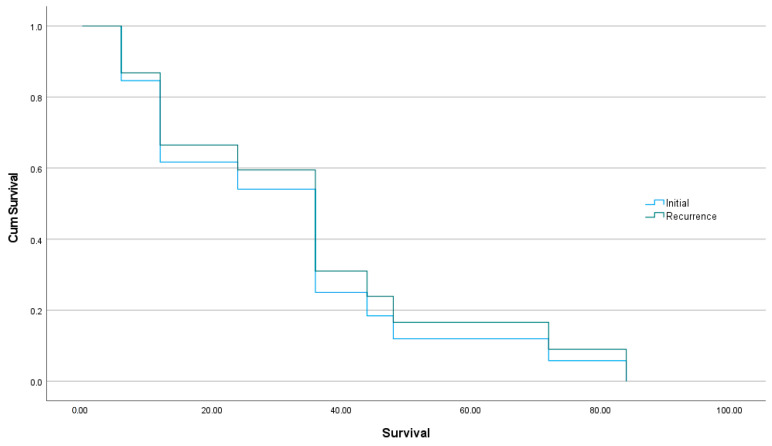
Survival curves for women with axillary lymph node metastases presenting at initial diagnosis or at recurrence.

**Figure 3 jpm-13-01532-f003:**
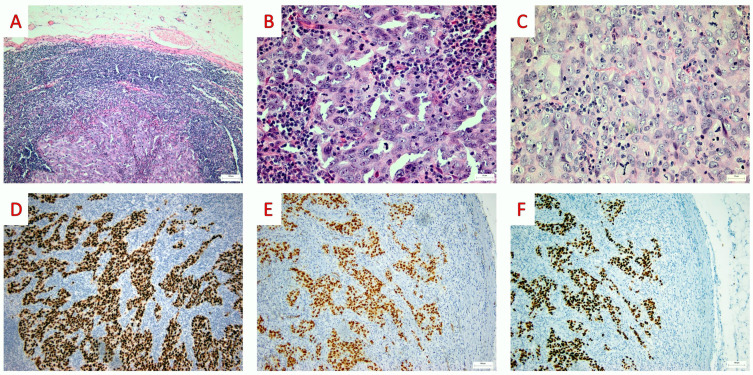
**Lymph node biopsy: histopathologic findings.** (**A**) On low-power examination, the lymph node is infiltrated by a poorly differentiated carcinoma (Hematoxylin and Eosin, H&E; ×100). (**B**,**C**) On higher power examination, the cells are pleomorphic and several mitotic figures are readily identifiable (H&E; ×400). (**D**–**F**) On immunohistochemical examination, the nucleus of the tumor cells stained for PAX-8 (PAX-8 mouse monoclonal MRQ-50, Roche; ×100), WT-1 (WT1 mouse monoclonal 6F-H2, DAKO; ×100), and p53 (mouse monoclonal DO7, DAKO; ×100) (original, previously unpublished photos).

**Table 1 jpm-13-01532-t001:** Clinical and demographic data of cases with axillary lymph node metastasis as initial presentation.

Authors	Year	Age	Laterality		Ovarian Tumor Size	CA-125 Level
			Ovary	Axillary LN		
Hockstein et al. [6]	1997	78	Bilateral	Right	6 cm	247 U/mL
Gallo et al. [7]	1997	59	Bilateral	Bilateral	NM	1024 U/mL
Moustafa and Beynon [9]	2002	50	Left	Left	5 cm	481 U/mL
Cecarelli et al. [14]	2010	48	Right	Right	3 cm	230 U/mL
Harrison et al. [15]	2010	74	Right	Right	4.1 cm	900 ng/mL
Sibio et al. [17]	2013	49	Bilateral	Right	NM	750 μm/mL
Choi et al. [18]	2014	68	Right	Left	NM	660.9 U/mL
Saxena et al. [20]	2014	45	Right	Right	11.4 cm	932 U/mL
Eitan et al. [22]	2017	70	Bilateral	Right	NM	4758
Eitan et al. [22]	2017	51	Bilateral	Left	NM	1675
Zuhdy et al. [23]	2019	39	Left	Right	8 cm	87
Zuhdy et al. [23]	2019	58	Right	Right	14 cm	720 U/mL
Mirzaei et al. [24]	2022	55	Bilateral	Left	NM	512 U/mL

LN: Lymph node; NM: not mentioned.

**Table 2 jpm-13-01532-t002:** Management data of cases with axillary lymph node metastasis as the initial presentation.

Authors	Surgical Management	Adjuvant Therapy	Recurrence	Outcome (Months)
Hockstein et al. [6]	BSO, OM, pelvic, and para-aortic LND	Tamoxifen, CP after recurrence	Yes *	ANED 24
Gallo et al. [7]	HBSO, OM, pelvic peritonectomy, PLNB, rectosigmoidectomy, and bilateral ALND	CEC	No	ANED 12
Moustafa and Beynon [9]	HBSO, OM, and para-aortic lymph node mass excision	CP	NM	NM
Cecarelli et al. [14]	HBSO, OM, peritoneal biopsies, pelvic peritonectomy, and pelvic and para-aortic LND	HIPEC, CP	No	ANED 48
Harrison et al. [15]	None	CP	NM	ANED
Sibio et al. [17]	HBSO, OM, appendectomy, peritonectomy, iliac, para-aortic, and inferior mesenteric artery LND, peritoneal implant ablation, and bladder peritoneum implant excision	HIPEC, CP	No	ANED 84
Choi et al. [18]	HBSO and pelvic LND	Neoadj. CP	NM	NM
Saxena et al. [20]	HBSO, peritoneal seedings, abdominal LNs, and ALND	CP	No	ANED 12
Eitan et al. [22]	HBSO, OM, and small intestine biopsies	Neoadj. CP, CPB	Yes **	AWD 6
Eitan et al. [22]	HBSO, OM, and umbilical biopsy	Neoadj. CP, CPB	Yes ***	AWD at the end of therapy
Zuhdy et al. [23]	HLSO, infracolic OM, peritoneal nodules excision, and ALND	NA	NM	Lost to follow-up
Zuhdy et al. [23]	HBSO, infracolic OM, and iliac LNs sampling	Neoadj. CP, CP	NM	Lost to follow-up
Mirzaei et al. [24]	HBSO	CPB	No	ANED 12

ALND: axillary lymph node dissection; ANED: alive with no evidence of disease; AWD: alive with disease; BSO: bilateral salpingo-oophorectomy; CEC: cisplatin, epirubicin, and cyclophosphamide; CP: carboplatin–paclitaxel; CPB: carboplatin–paclitaxel–bevacizumab; HBSO hysterectomy + bilateral salpingo-oophorectomy; HIPEC: hyperthermic intraperitoneal chemotherapy; HLSO: hysterectomy + left salpingo-oophorectomy; LN: lymph node; LND: lymph node dissection; NA: unavailable data; Neoadj: neoadjuvant; NM: not mentioned; OM: omentectomy; PLNB: pelvic lymph node biopsy; *: recurrence at the sigmoid colon after 16 months, treated with low anterior resection; **: recurrence at mediastinum, few peritoneal implants, retroperitoneal and bilateral axillary lymph nodes; and ***: recurrence as pelvic mass and peritoneal carcinomatosis.

**Table 3 jpm-13-01532-t003:** Clinical and demographic data of cases with axillary lymph node metastasis as recurrence.

Authors	Year	Age	Laterality		Ovarian Tumor Size	CA-125 Level	Stage
			Ovary	Axillary LN			
Orris et al. [8]	1999	63	NM	Bilateral	NM	NM	IIIc
Ozmen et al. [10]	2007	74	NM	Right	2 cm	NM	IIIa
Ozmen et al. [10]	2007	38	ΝΜ	Right	3 cm	98 ng/mL	IIIa
Skagias et al. [11]	2008	63	NM	Right	NM	NM	IIIb
Woo and Kim [12]	2008	43	NM	Right	NM	437 U/mL	IIIc
Aydin et al. [13]	2009	47	Right	Right	25 cm	600 μm/mL	IIIc
Goyal et al. [16]	2012	68	NM	Left	NM	104 U/mL	IIIc
Patel et al. [19]	2014	50	NM	Bilateral	NM	204 U/mL	IIIc
Nazos et al. [21]	2107	79	Right	Right	NM	51 U/mL	IIIc
Zuhdy et al. [23]	2019	69	Bilateral	Right	NM	NM	IIIb
Phung et al. [25]	2022	58	NM	Left	NM	41.2 U/l	IIIc
Hobek and Onan [26]	2023	37	Bilateral	Left	8.2/8.5 cm	26.7 U/mL	IIIc

LN: Lymph node; NM: not mentioned.

**Table 4 jpm-13-01532-t004:** Management data of cases with axillary lymph node metastasis as recurrence.

Authors	Surgical Management	Adjuvant Therapy	Surgical Management of Recurrence	Adjuvant Therapy for Recurrence	Interval to Metastasis	Outcome
Orris et al. [8]	HBSO, OM, peritonectomy, and sigmoidectomy	CP, IPC	Bilateral ALND	CP	37 months	NM
Ozmen et al. [10]	HBSO, OM, and pelvic LND	CP	ALND, LAR	CP	48 months	ANED 72
Ozmen et al. [10]	HBSO, OM, and pelvic LND	Neoadj. CP, CP	ALND	ACT NS	24 months	ANED 36
Skagias et al. [11]	Cytoreductive surgery NS	ACT NS	NM	NM	NM	NM
Woo and Kim [12]	HBSO, OM, and pelvic LND	CP, CPG (*)	ALND	NM	28 months (**)	NM
Aydin et al. [13]	HBSO, OM, peritoneal biopsies, and pelvic LND	CP	ALND	CP	32 months	ANED 44
Goyal et al. [16]	Cytoreductive surgery NS	CP	ALND	NM	NM	ANED 36
Patel et al. [19]	Debulking surgery NS	Neoadj. CP, CP	NM	NM	25 months	NM
Nazos et al. [21]	HBSO, OM, pelvic LND appendectomy, and peritoneal and diaphragmatic biopsies	CP	None	RT	24 months	AWD 36
Zuhdy et al. [23]	HBSO, iliac LN sampling, and infracolic OM	Neoadj. CP, CP	None	CP, (***) CP, GP, AI	28 months	AWD 36
Phung et al. [25]	Debulking surgery NS	CP	None	CP	12 months	ANED 16
Hobek and Onan [26]	HBSO, OM, appendectomy, and pelvic-paraaortic LND LAR	CP	None	CP	6 months	NM

ACT: adjuvant chemotherapy; AI: aromatase inhibitor; ALND: axillary lymph node dissection; BSO: bilateral salpingo-oophorectomy; CP: carboplatin–paclitaxel; CPB: carboplatin–paclitaxel–bevacizumab; CPG: carboplatin–paclitaxel–gemcitabine; GC: gemcitabine–paclitaxel; HBSO: hysterectomy + bilateral salpingo-oophorectomy; HIPEC: hyperthermic intraperitoneal chemotherapy; HLSO: hysterectomy + left salpingo-oophorectomy; IPC: intraperitoneal cisplatin; LAR: low anterior resection; LND: lymph node dissection; Neoadj: neoadjuvant; NS: not specified; OM: omentectomy; RT: radiotherapy; *: administered due to disease recurrence 45 months after postoperatively; **: calculated after disease recurrence; and ***: treatment after second recurrence in the peritoneum.

## Data Availability

Not applicable.

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
