# Peer review of "Axillary Lymph Node Metastasis from Ovarian Carcinoma: A Systematic Review of the Literature"

_jpm, 2023, doi:10.3390/jpm13111532_

Round 1

Reviewer 1 Report

Comments and Suggestions for Authors

It is well written manuscript on the topic that is rather seldom. PRISMA guidelines are followed what is worthy in this kind of manuscripts.

One major issue that should be corrected is that not only in the abstract section but also in the section "conclusions" there are no conclusions. I suggest rewrite text in this two sections with new statements of conclusions

Author Response

It is well written manuscript on the topic that is rather seldom. PRISMA guidelines are followed what is worthy in this kind of manuscripts.

One major issue that should be corrected is that not only in the abstract section but also in the section "conclusions" there are no conclusions. I suggest rewrite text in this two sections with new statements of conclusions

Answer:

Dear reviewer,

Thank you for your comments. We have modified the abstract section and the section of conclusions with new statements.

Reviewer 2 Report

Comments and Suggestions for Authors

Dear authors

Excellent working. However, if you give more information about the start date of the journal that you were searching, it will be perfect.

Best Regards

Author Response

Dear authors

Excellent working. However, if you give more information about the start date of the journal that you were searching, it will be perfect.

Best Regards

Answer:

Dear reviewer,

Thank you for your comments. We did not specify start date during search. However, the publication of the first two cases of ovarian carcinoma axillary metastasis was in 1997. We have modified the manuscript to highlight this.

Reviewer 3 Report

Comments and Suggestions for Authors

The authors present the article "Axillary Lymph Node Metastasis from Ovarian Carcinoma: A Systematic Review of the Literature", which is an interesting proposal but somewhat limited by the articles reviewed for the review.

The main limitation of the study is the number of articles reviewed, in fact the authors mention it as one of its limitations.

I consider that the authors should mention in the introduction the prevalence of metastasis to axillary lymph nodes or the percentage of ovarian cancer cases that present this type of metastasis, to establish the importance of the study.

In their discussion they should discuss why the frequency of metastasis to axillary nodules is low or why they consider that there are few reports.

Comments on the Quality of English Language

Minor editing of English language required

Author Response

Reviewer nr.3

The authors present the article "Axillary Lymph Node Metastasis from Ovarian Carcinoma: A Systematic Review of the Literature", which is an interesting proposal but somewhat limited by the articles reviewed for the review.

The main limitation of the study is the number of articles reviewed, in fact the authors mention it as one of its limitations.

I consider that the authors should mention in the introduction the prevalence of metastasis to axillary lymph nodes or the percentage of ovarian cancer cases that present this type of metastasis, to establish the importance of the study.

In their discussion they should discuss why the frequency of metastasis to axillary nodules is low or why they consider that there are few reports.

Answer:

Dear reviewer,

Thank you for your comments. We have added to the introduction information regarding the scarcity of metastasis to axillary lymph nodes, unfortunately, there is no literature data for the prevalence. In the discussion section, we have added a comment regarding the low frequency of axillary lymph node metastasis

Reviewer 4 Report

Comments and Suggestions for Authors

The authors conducted a systematic review on axillary lymph node metastasis from ovarian carcinoma which is a very rare condition.

Authors found 25 cases of ovarian carcinoma with axillary metastasis. The paper followed the PRISMA and PICO guidelines. The data is well described and analyzed. The results are well presented with a good scientific language.

Author Response

Reviewer nr.4

The authors conducted a systematic review on axillary lymph node metastasis from ovarian carcinoma which is a very rare condition.

Authors found 25 cases of ovarian carcinoma with axillary metastasis. The paper followed the PRISMA and PICO guidelines. The data is well described and analyzed. The results are well presented with a good scientific language.

Answer:

Dear reviewer,

Thank you for your comments and for providing your expert opinion in this review.